# Tunicamycin as a Novel Redifferentiation Agent in Radioiodine Therapy for Anaplastic Thyroid Cancer

**DOI:** 10.3390/ijms22031077

**Published:** 2021-01-22

**Authors:** Yoon Ju Choi, Jae-Eon Lee, Hyun Dong Ji, Bo-Ra Lee, Sang Bong Lee, Kil Soo Kim, In-Kyu Lee, Jungwook Chin, Sung Jin Cho, Jaetae Lee, Sang-Woo Lee, Jeoung-Hee Ha, Yong Hyun Jeon

**Affiliations:** 1Department of Nuclear Medicine, School of Medicine, Kyungpook National University, Daegu 41566, Korea; hongsi1221@naver.com (Y.J.C.); jihd0210@naver.com (H.D.J.); jaetae@knu.ac.kr (J.L.); 2Laboratory Animal Center, Daegu-Gyeongbuk Medical Innovation Foundation, Daegu 41404, Korea; koof12@naver.com (J.-E.L.); damsmom@dgmif.re.kr (B.-R.L.); kslac@dgmif.re.kr (K.S.K.); 3Vaccine Commerialization Center, Gyeongbuk Institute for Bioindustry, 88, Saneodanjigil, Pungsan-eup, Andong-si, Gyeongbuk 36618, Korea; sangbongyi1@gmail.com; 4College of Veterinary Medicine, Kyungpook National University, Daegu 41566, Korea; 5Department of Internal Medicine, School of Medicine, Kyungpook National University, Daegu 41944, Korea; leei@knu.ac.kr; 6New Drug Development Center, Daegu-Gyeongbuk Medical Innovation Foundation, Daegu 41404, Korea; jwchin@dgmif.re.kr (J.C.); sjcho@dgmif.re.kr (S.J.C.); 7Department of pharmacology, School of Medicine, Kyungpook National University, Daegu 41405, Korea; 8Leading-Edge Research Center for Drug Discovery and Development for Diabetes and Metabolic Disease, Kyungpook National University Hospital, Daegu 41404, Korea; 9Research Institute of Aging and Metabolism, Kyungpook National University, Daegu 41404, Korea

**Keywords:** tunicamycin, sodium-iodide symporter, anaplastic thyroid cancer, redifferentiation, radioiodine therapy

## Abstract

The silencing of thyroid-related genes presents difficulties in radioiodine therapy for anaplastic thyroid cancers (ATCs). Tunicamycin (TM), an N-linked glycosylation inhibitor, is an anticancer drug. Herein, we investigated TM-induced restoration of responsiveness to radioiodine therapy in radioiodine refractory ATCs. ^125^I uptake increased in TM-treated ATC cell lines, including BHT101 and CAL62, which was inhibited by KClO_4_, a sodium-iodide symporter (NIS) inhibitor. TM upregulated the mRNA expression of iodide-handling genes and the protein expression of NIS. TM blocked pERK1/2 phosphorylation in both cell lines, but AKT (protein kinase B) phosphorylation was only observed in CAL62 cells. The downregulation of glucose transporter 1 protein was confirmed in TM-treated cells, with a significant reduction in ^18^F-fluorodeoxyglucose (FDG) uptake. A significant reduction in colony-forming ability and marked tumor growth inhibition were observed in the combination group. TM was revealed to possess a novel function as a redifferentiation inducer in ATC as it induces the restoration of iodide-handling gene expression and radioiodine avidity, thereby facilitating effective radioiodine therapy.

## 1. Introduction

Well-differentiated thyroid cancer exhibits active radioactive iodide uptake owing to the appropriate expression of iodide-handling genes, including sodium-iodide symporter (NIS), thyroid peroxidase (TPO), and thyroid-stimulating hormone receptor (TSHR). This facilitates the uptake and organification of radioactive iodine, resulting in good therapeutic outcomes following radioiodine therapy [1,2]. The genetic alteration of serine/threonine-protein kinase B-Raf (BRAF)^V600E^, Neurotrophic Receptor Tyrosine Kinase 1 (NTRK1), rearranged during transfection (RET), and the three isoforms of RAS (N, H, and K) occurs in anaplastic thyroid carcinoma (ATC) with aggressive metastasis to the lungs, bones, and regional lymph nodes. These events cause constitutive mitogen-activated protein kinase (MAPK) signaling, which induces the continuous suppression or silencing of iodide-handling genes, leading to decreased radioiodine avidity and, finally, the failure of radioiodine therapy. Many studies have been conducted to develop methods to restore radioiodine avidity by NIS delivery, epigenetic modulation, and MAPK inhibition in PD/ATC cells. However, most of these approaches have not produced satisfactory clinical outcomes. Accordingly, new approaches are urgently needed to restore radioiodine avidity via redifferentiation induction with high-efficiency.

It has been reported that N-linked glycosylation may play an important role in maintaining protein stability, localization, and function [3]. Many studies have shown that tunicamycin (TM), a naturally occurring antibiotic, can inhibit N-linked glycosylation in several types of cancer cells, presenting a useful therapeutic approach for cancer therapy by decreasing angiogenesis [4], inducing endoplasmic reticulum stress-induced autophagy and apoptosis, and increasing TRAIL-induced cell death (apoptosis) [5,6,7]. Another report demonstrated that TM-induced N-linked glycosylation inhibition increases sensitivity to cisplatin-mediated cytotoxicity in head and neck cancer cells [8]. Even though several interesting findings have indicated the promising anticancer effects of TM, the effect of TM on redifferentiation has not been explored in advanced thyroid cancer cells, especially in ATC cells with poor differentiation.

Herein, we explored the effects of TM on iodide- and glucose-handling gene expression in ATC cells with a KRAS or BRAF mutation. Furthermore, signal transduction (especially levels of phosphorylated AKT and ERK) and the functional activity of NIS and a glucose transporter was determined using radioiodine and ^18^F-fluorodeoxyglucose (FDG) uptake assays. Finally, the applicability of TM for ATC treatment was evaluated in vitro and in vivo.

## 2. Results

### 2.1. Restoration of Radioiodine Avidity in ATC Cells by TM

The effects of TM on the proliferation of CAL62/effluc and BHT101/effluc cells were examined using in vitro bioluminescence imaging (BLI) 24 h and 48 h after treatment. The proliferation of both ATC cell lines was reduced in a dose- and time-dependent manner (Appendix A).

Next, we determined whether TM could induce the restoration of radioiodine avidity in CAL62 and BHT101 cells. After treatment with various doses of TM, we conducted radioiodine uptake assays. TM significantly increased radioiodine uptake in both ATC cell lines in a dose-dependent manner (Figure 1A,B). The maximum relative increase in iodide uptake following treatment with 2 µM TM was 9.4- and 3.6-fold in CAL62/effluc cells and BHT101/effluc cells, respectively, compared to the vehicle group (dimethyl sulfoxide treated). To determine whether the increased radioiodine avidity was associated with TM-mediated NIS function, we performed an inhibition study with KClO_4_, a specific inhibitor of NIS. The enhanced radioiodine avidity was completely inhibited by KClO_4_ in TM-treated CAL62/effluc and BHT101/effluc cells (Figure 1C,D), indicating that the restoration of radioiodine avidity was related to the change in TM-induced iodide-handling genes associated with NIS function.

### 2.2. Restoration of Iodide-Handling Genes by TM in ATC Cells

We examined the effect of TM on iodide-handling genes such as *NIS*, *TPO*, thyroglobulin (*TG*), and *TSHR* in both ATC cell lines using quantitative real-time PCR and Western blotting analysis. Quantitative real-time PCR revealed the restoration of TM-mediated iodide-handling gene expression in CAL62 and BHT101 cells (Figure 2). The mRNA expression levels of NIS, TPO, and TG in both ATC cell lines reached a peak upon treatment with TM as low as 0.5 μM. The maximum expression of *TSHR* mRNA in CAL62 cells was detectable at 2 μM TM. Immunoblotting analysis showed that TM induced a marked increase in iodide-handling gene expression in CAL62 and BHT101 cells (Figure 3A,B). The expression of *NIS* was the highest in CAL62 and BHT101 cells in the presence of 0.5 μM TM. For other iodide-handling genes such as *TPO*, *TG*, and *THSR*, the peak expression was observed upon treatment with 1~2 μM TM. We observed an increase in membrane NIS protein in both ATC cell lines, as determined by biotinylation cell surface protein isolation assays (Figure 3C).

### 2.3. Regulation of MAPK and AKT, As Well As Glucose Metabolism, by TM in ATC Cells

The MAPK and AKT pathways were examined following TM treatment in both ATC cell lines. TM led to ERK phosphorylation inhibition in CAL62 and BHT101 cells (Figure 4A,B). However, the downregulation of phosphorylated AKT was only detectable in TM-treated CAL62 cells. Next, we determined whether TM affected glucose metabolism as well as glucose transporter expression. The significant downregulation of Glut1 was observed in TM-treated CAL62 and BHT101 cells (Figure 4A,B). Consistently, a marked reduction in ^18^F-FDG uptake was observed in both ATC cell lines, with a 44% and 25% relative reduction in glucose uptake in CAL62 cells and BHT101 cells, respectively (Figure 4C).

### 2.4. Restoration of Unresponsiveness to Radioiodine Therapy by TM in Vitro and in Vivo

We explored whether TM can reverse the unresponsiveness of radioiodine therapy against radioiodine refractory ATC. Treatment with either ^131^I or TM alone failed to decrease colony formation ability in BHT101 cells (Figure 5A). However, the combination treatment of TM and ^131^I resulted in a drastic reduction in colony formation.

We additionally conducted a biodistribution study to evaluate radioiodine uptake in BHT101 tumors. As shown in Appendix A, 0.02 mg/kg TM induced increased radioiodine avidity in BHT101 tumors. However, there was no increase in radioiodine avidity in vehicle-treated mice. Radioiodine uptake was also seen in the thyroid and stomach due to physiological reasons.

The potential of combination therapy of TM and ^131^I was evaluated in ATC-tumor-bearing mice, as illustrated in Figure 5B. Rapid tumor progression was observed in vehicle-treated mice (Figure 5C). Similarly, mice treated with either TM or ^131^I alone showed aggressive tumor growth during therapy monitoring. Treatment with TM and I-131 led to significant retardation of tumor growth. Bodyweight reduction and abnormal behavior were not observed in all mice during in vivo therapy (data not shown).

## 3. Discussion

Thyroidectomy is the primary treatment for well-differentiated thyroid cancer, followed by conventional radioiodine therapy for the complete removal of remaining tumor lesions [9]. The full expression of iodide-handling genes is a key factor for successful radioiodine therapy in thyroid cancer. However, poor differentiation in thyroid cancer leads to the downregulation of iodide-handling genes and loss of radioiodine avidity, both of which are found in ATC. Genetic studies have demonstrated that ATC originates from pre-existing papillary thyroid cancer involving *BRAF* or *RAS* mutations that have subsequently evolved toward ATC by acquiring additional genetic aberrations, particularly in *TP53*, *PIK3CA*, and *TERT* promoters. Although many attempts were made to restore radioiodine avidity in advanced thyroid cancer, an effective approach has not been found.

Several studies have demonstrated that TM can induce anticancer effects via the inhibition of vascular formation, autophagy induction, and sensitization to anticancer agents such as TRAIL, cisplatin, and trastuzumab in various types of cancer [5,8,10]. Even though several interesting reports have indicated the promising anticancer effects of TM, the effect of TM on redifferentiation has not been explored in advanced thyroid cancer cells, especially in ATC cells with poor differentiation.

To investigate whether TM has the potential as an inducer to restore iodide-handling gene expression as well as radioiodine avidity in ATC cells, we evaluated its effects on radioiodine uptake in CAL62 cells and BHT101 cells carrying *KRAS* and *BRAF* gene mutations, respectively. These cells have been used to determine the therapeutic effects of several drugs associated with cytotoxicity and iodide-handling gene restoration [11,12,13]. TM induced an increase in radioiodine uptake in a dose-dependent manner, which was completely inhibited by the NIS-specific inhibitor KClO_4_. Other previous literature [14,15] have demonstrated that tyrosine kinase inhibitors (cf. selumetinib) and retinoic acid can re-differentiate thyroid cancer and non-thyroidal cancer, leading to the restoration of radioiodine avidity. However, when these agents were applied to anaplastic thyroid cancer cells, they were not effective in restoring radioiodine avidity and iodide-handling gene expression (Appendix A). These findings suggest that the restoration of radioiodine avidity and the restoration of iodide-handling genes in anaplastic thyroid cancer are related to TM-modulated NIS functioning.

A unique property of thyroid follicular cells is the ability to trap and concentrate iodide, which depends on the expression of NIS, TG, TPO, and TSHR. The NIS protein mediates iodide uptake in normal and well-differentiated thyroid cancer. TPO iodinates TG, which leads to iodide retention within thyroid follicles. Restoration of the expression of these iodide metabolism-related genes is associated with successful radioiodine therapy in ATC. In our study, the protein levels of iodide-handling genes were downregulated in CAL62 and BHT101 cells, consistent with previous clinical findings [16]. However, TM could simultaneously restore NIS, as well as TPO, TG, and TSHR mRNA and protein expression levels in CAL62 cells and BHT101 cells.

Clinical evidence indicates that ATC exhibits not only a high FDG uptake with increased glucose transporter expression levels but also a marked reduction in radioiodine avidity [17,18,19]. Feine et al. also stated that increased glucose metabolism is a sign of greater malignancy [20]. Our results showed that TM treatment led to the reduction of Glut1 and glucose metabolism. Taken together, the restoration of iodide-handling genes and downregulation of glucose metabolism indicate that TM can re-induce the differentiation of ATC cells.

Constitutive MAPK and AKT activation is related to ATC, resulting in poor responsiveness to radioiodine therapy. Thus, many attempts have been made to downregulate MAPK and AKT signaling to restore iodide avidity using various kinase inhibitors such as cabozantinib, dabrafenib, sorafenib, LY294002, and selumetinib in preclinical and clinical situations [21,22,23,24,25,26]. When we examined TM-induced signal transduction in both ATC cell lines, it was found that TM also induced the significant reduction of ERK phosphorylation (CAL62 and BHT101 cells) and AKT phosphorylation (CAL62 cells), indicating that the restoration of iodide-handling genes and iodide avidity may be related to the downregulation of the MAPK and AKT pathway.

Successful radioiodine therapy depends on the effective incorporation of therapeutic radioiodine-131 into the cancerous thyroid gland. Based on our finding of the TM-mediated increase in radioiodine avidity in vitro, we presumed that TM-induced redifferentiation reduces unresponsiveness to radioiodine therapy in ATC. As expected, we observed a drastic reduction in colony-forming ability when a combined treatment of TM and ^131^I was introduced to ATC cells. Consistent with the in vitro findings, the combination of TM and ^131^I led to marked retardation of tumor growth in vivo, whereas few therapeutic effects were observed in vehicle-treated, TM-treated, or ^131^I-treated mice. These data indicate that the TM-modulated upregulation of iodide-handling genes is a reasonable strategy for the induction of effective radioiodine therapy.

Increasing iodine avidity is very important in differentiated thyroid cancer (DTC) as well as ATC because positive iodine uptake does not always result in a good therapeutic response in patients with DTC. Our findings would be better if TM induced the change of iodide handling genes and radioiodine avidity in the DTC cells. When we have examined the effects of TM on radioiodine uptake, as well as iodide-handling genes expression of BCPAP as differentiated papillary thyroid cancer, TM led to an increase in radioiodine avidity in BCPAP cells (Appendix A). KCLO4, as a NIS-specific inhibitor, completely reduced the enhanced radioiodine uptake to the basal level (Appendix A). Furthermore, Western blotting analysis showed the TM-induced upregulation of NIS, TPO, and TSHR protein expression (Appendix A). These results suggested that TM is also an effective agent for restoring radioiodine avidity in differentiated thyroid cancer.

Despite our interesting finding that TM restored iodide-handling genes in ATC, some issues should be addressed. Additional in vivo studies are warranted to conclusively establish their responsiveness to radioiodine therapy in other types of ATC models. Regarding signal transduction, the detailed mechanism of the TM-induced recovery of radioiodine avidity was not addressed; further studies should be conducted in this direction.

In conclusion, in this study, we demonstrated the ability of TM, a well-known N-linked glycosylation inhibitor, to re-differentiate ATC cells, thereby effectively restoring iodide-handling gene expression and radioiodine avidity. TM also downregulated glucose metabolism, as well as MAPK and AKT expression (Figure 6). Importantly, the TM-mediated redifferentiation of ATC allows for successful radioiodine therapy in radioiodine therapy-refractive ATC tumors. Our findings reveal the utility of TM as a re-differentiating agent for previously developed drugs (likely drug-repositioning candidates) in ATC.

## 4. Materials and Methods

### 4.1. Cells

Two ATC cell lines, CAL62 and BHT-101, were purchased from Deutsche Sammlung von Mikrooganismen und Zellkulturen. Both cell lines were maintained in high-glucose Dulbecco’s modified Eagle medium (Hyclone; Thermo Fisher Scientific, Waltham, MA, USA), supplemented with 10% fetal bovine serum (Hyclone) and a 1% antibiotic-antimycotic (Hyclone) at 37 °C in a 5% CO_2_ atmosphere. CAL62 and BHT101 cells were transduced with a retroviral vector containing enhanced firefly luciferase and Thy1.1 genes. After infection, cells were stained with anti-Thy1.1 antibody and sorted using FACSAria (BD Biosciences, San Jose, CA, USA). Sorted cells were enriched and called CAL62/effluc and BHT101/effluc.

### 4.2. ^125^I Uptake Assay

Cells were plated in 24-well plates (Thermo Fisher Scientific, Waltham, MA, USA) for 24 h, treated with tunicamycin (Sigma, St. Louis, MO, USA) as a 10 mM stock solution in dimethyl sulfoxide, and stored at 36 °C for 24 h. After aspiration of the drug-containing medium, the cells were washed with 1 mL of Hanks balanced salt solution (HBSS) (Thermo Fisher Scientific) and incubated with 500 mL of HBSS containing 0.5% bovine serum albumin (Sigma) (bHBSS), 3.7 kBq of carrier-free ^125^I (PerkinElmer, MA, USA), and a 10 µmol/L solution of sodium iodide (specific activity of 740 MBq/mmol) at 37 °C for 30 min. The cells were then washed twice with ice-cold bHBSS and lysed with 500 mL of 2% sodium dodecyl sulfate (Sigma). The radioactivity was measured using a gamma counter (Cobra II; Canberra Packard, Packard Bioscience; PerkinElmer). The radioactivity of the cells was normalized using total protein concentrations determined by a bicinchoninic acid protein assay kit (Pierce Protein Biology; Thermo Fisher Scientific). Some cells were preincubated with 300 μM KClO_4_ (a specific inhibitor for NIS) (Sigma) for 30 min to inhibit iodide uptake, followed by the ^125^I uptake test.

### 4.3. ^18^F-FDG Uptake Assay

The change in ^18^F-FDG uptake was evaluated as described previously [27], and detailed information is described in the Appendix A.

### 4.4. Quantitative RT–PCR

The change in iodide-handling gene expression was evaluated as described previously [27], and detailed information is described in the Appendix A.

### 4.5. Western Blot

The change in protein expression was evaluated as described previously [27], and detailed information is described in the Appendix A.

### 4.6. Clonogenic Assay

Cells were seeded on a T-75 Flask (Thermo Fisher Scientific) and treated with 2 μM TM for 24 h. After 24 h, the cells were exposed to 1 mCi ^131^I (Korea Institute of Radiological and Medical Sciences, Seoul, Korea), incubated for 8 h, and 1000 cells per well were re-seeded in 6-well plates for 14 days. Finally, cells were fixed in 4% paraformaldehyde solution (Sigma) and stained with 0.05% crystal violet (Sigma).

### 4.7. Animals

Specific pathogen-free, 6-week-old, female BALB/c nude mice were obtained from SLC, Inc. (Shizuoka, Japan). All animals were maintained and used in accordance with The Guidelines for the Care and Use of Laboratory Animals of the Institute of Laboratory Animal Center, Daegu-Gyeongbuk Medical Innovation Foundation. The animal studies were conducted after approval by the Institutional Review Board on the Ethics of Animal Experiments of the Daegu-Gyeongbuk Medical Innovation Foundation (approval number: DGMIF-17120802-00, 8 December 2017).

### 4.8. In Vivo Therapy

To establish the ATC xenograft model, nude mice were subcutaneously challenged with 5 million BHT101 cells. When tumor size reached within 150 mm^3^, tumor-bearing mice were divided into four groups (*n* = 7) as follows: group 1, vehicle; group 2, 0.02 mg/kg TM (intraperitoneal injection) twice a week for 14 days; group 3, a single dose of 1 mCi ^131^I (intravenous injection); and group 4, 0.02 mg/kg TM twice a week for 14 days followed by a single dose of 1 mCi ^131^I. After the final TM treatment, tumor-bearing mice received radioactive ^131^I. Mouse body weight was monitored. Tumor size was measured with a caliper at the indicated time points, and tumor volume (mm^3^) was calculated using the following formula, tumor volume (mm^3^) = d^2^ × D/2, where d and D are the shortest and longest diameter in mm, respectively.

### 4.9. Statistical Analysis

All data are expressed as mean ± SD, and statistical significance was determined using an unpaired Student’s *t*-test in Prism 5 (GraphPad). *p*-values less than 0.05 were considered statistically significant.

## Figures and Tables

**Figure 1 ijms-22-01077-f001:**
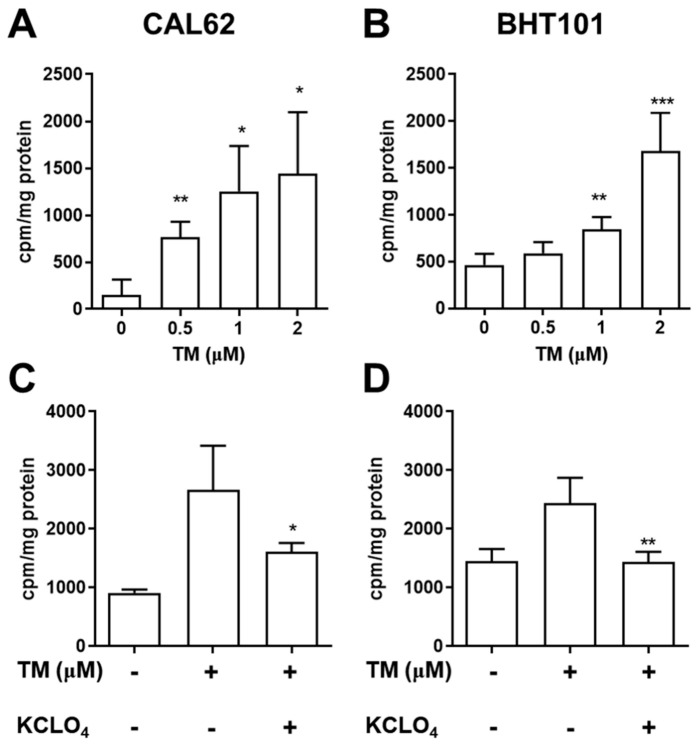
Effect of tunicamycin (TM) on iodide uptake in ATC cells. (**A**,**C**) CAL62 and (**B**,**D**) BHT101 cells were treated with various TM doses for 24 h, followed by iodide uptake assays. To examine the inhibition of iodide uptake, ATC cells were treated with 2 µM TM and 300 µM KClO_4_ (sodium-iodide symporter (NIS)-specific inhibitor) for 24 h, and iodide uptake was assessed. * *p* < 0.05, ** *p* < 0.005, *** *p* < 0.0005, compared to absence of TM (Iodide uptake assay). * *p* < 0.05, ** *p* < 0.005 compared with TM (KCLO_4_ inhibition study). Data are the mean ± SD of three samples per group.

**Figure 2 ijms-22-01077-f002:**
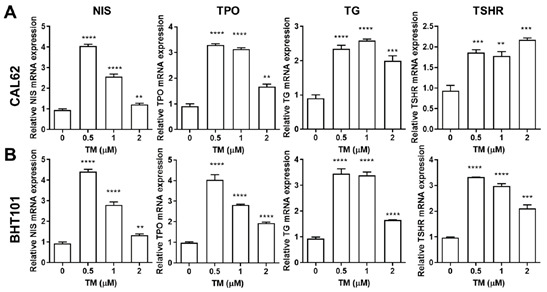
Upregulated mRNA expression of iodide-handling genes. Relative increases in iodide-handling gene expression in TM-treated (**A**) CAL62 and (**B**) BHT101 cells 24 h post-treatment. ** *p* < 0.005, *** *p* < 0.0005, **** *p* < 0.0001 compared to absence of tunicamycin. Data are the mean ± SD of three samples per group.

**Figure 3 ijms-22-01077-f003:**
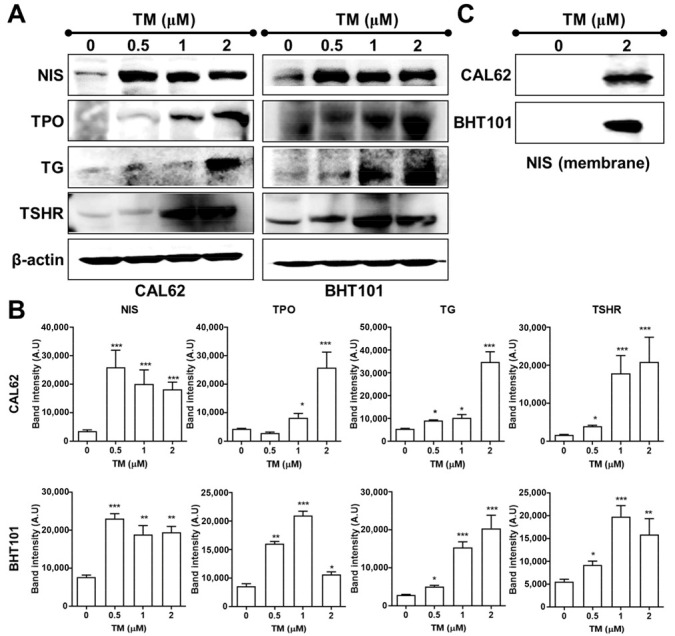
Restoration of iodide-handling genes by tunicamycin (TM). (**A**) Immunoblotting analysis of iodide-handling gene expression in CAL62 and BHT101 cells treated with different doses of TM at 24 h post-treatment. (**B**) Quantitative analysis of respective iodide-handling gene levels by scanning densitometry. (**C**) Immunoblotting analysis of membrane NIS protein levels in CAL62 and BHT101 cells treated with 2 µM TM at 24 h post-treatment. The membrane fraction was collected by previously described biotinylation-mediated cell surface protein isolation methods. * *p* < 0.05, ** *p* < 0.005, *** *p* < 0.0005, compared to absence of TM. Data are the mean ± SD of three samples per group.

**Figure 4 ijms-22-01077-f004:**
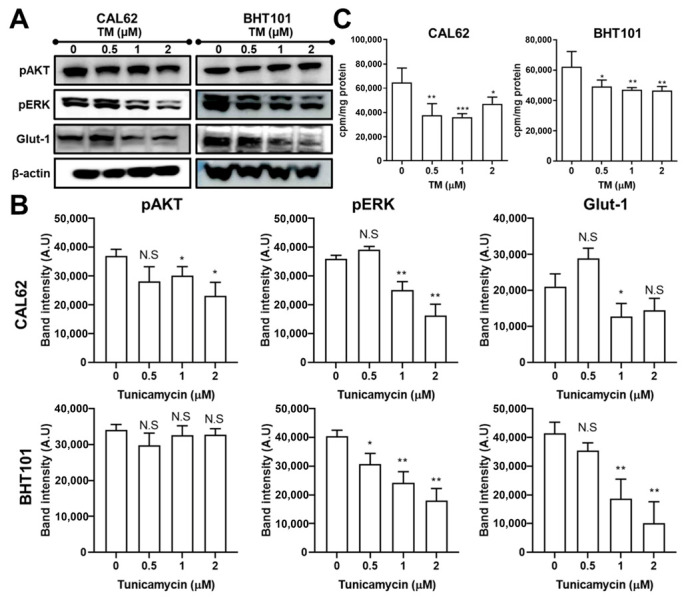
Inhibition of the MAPK pathway and glucose metabolism by tunicamycin (TM). (**A**,**B**) Immunoblotting analysis of AKT and ERK phosphorylation levels and Glut1 protein level in CAL62 and BHT101 cells treated with 2 µM TM at 24 h post-treatment. Quantitative analysis of pAKT, pERK, and Glut1 levels by scanning densitometry. (**C**) ^18^F-fluorodeoxyglucose (FDG) uptake levels in TM-treated CAL62 and BHT101 cells at 24 h post-treatment. CPM, count per minute. * *p* < 0.05, ** *p* < 0.005, and *** *p* < 0.0005 compared to absence of TM. Data are the mean ± SD of three samples per group.

**Figure 5 ijms-22-01077-f005:**
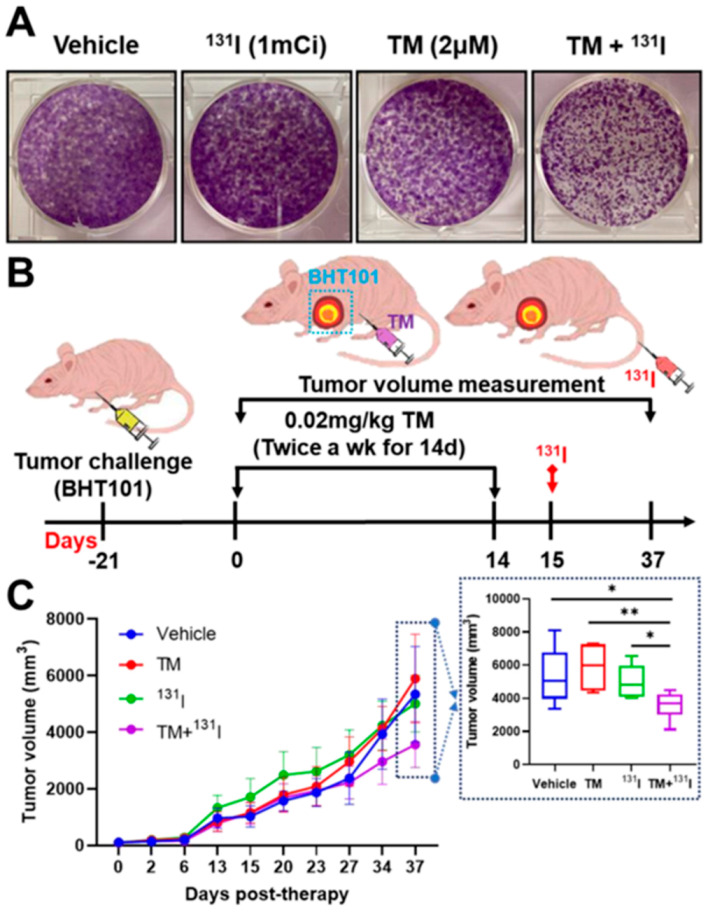
Evaluation of therapeutic response to ^131^I therapy with TM treatment**.** (**A**) Colony formation assay. BHT101 cells treated with 2 μM TM and 1 mCi ^131^I were seeded onto 6-well plates, followed by further incubation for 14 days. After fixation, the cells were stained with crystal violet solution. (**B**) Brief scheme for in vivo therapy. When tumor mass was detectable by palpation and inspection, BHT101 tumor-bearing mice received 0.02 mg/kg TM via intraperitoneal injection twice a week for 2 weeks, followed by 1 mCi ^131^I treatment intravenously. Tumor volume was measured with a caliper at indicated times. (**C**) Tumor volume measurement. * *p* < 0.05, ** *p* < 0.005, compared with vehicle, TM, and ^131^I, respectively. Data are the mean ± SD of 7 or 8 mice.

**Figure 6 ijms-22-01077-f006:**
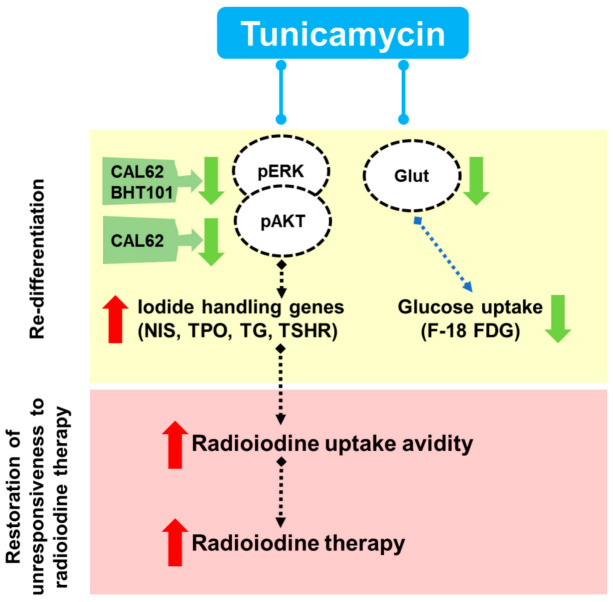
The proposed mode of action of TM-induced modulation of NIS function in anaplastic thyroid cancer (ATC) cells. TM restored the iodide-handling genes such as NIS, thyroid peroxidase (TPO), thyroglobulin (TG), and -stimulating hormone receptor (TSHR).

## Data Availability

Not applicable.

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
