# Peer review of "Tunicamycin as a Novel Redifferentiation Agent in Radioiodine Therapy for Anaplastic Thyroid Cancer"

_ijms, 2021, doi:10.3390/ijms22031077_

Round 1

Reviewer 1 Report

The title reflects the subject of the study. This manuscript presents a clear and clinically useful message. It is well written in terms of clarity, style, and use of English language. Materials and methods are sufficiently detailed. The discussion section explains adequately the purpose of this study in the context of published information. The conclusions accurately and clearly explain the main results. The length of the manuscript is ideal. All figures are of good quality and relevant to the subject. All references are appropriate and current.

Author Response

We appreciate your valuable comments on our research.

Reviewer 2 Report

This study is interesting and informative. However, several issues should be solved.

Major comments

  • To evaluate iodine-avidity after tunicamycin (TM) injection or the relationship between iodine uptake and therapeutic effects, iodine uptake (e.g., scan) in the tumor model should be compared between two groups (with or without TM injection). Or autoradiography might be possible if scan images were not available.
  • Increasing iodine avidity is very important in differentiated thyroid cancer (DTC) as well as ATC, because positive iodine uptake does not always result in a good therapeutic response in patients with DTC. Your study would be better if you showed the change of iodine uptake after TM injection and related molecular profiles in the DTC cell line and tumor model.
  • Several drugs or agents (e.g., TKI, retinoic acid) have been studied for redifferentiation of thyroid cancer cells. So, you had better describe the effect of those agents in terms of redifferentiation with detailed manners and, if possible, compare your results with previous reports based on the degree of redifferentiation in the discussion section.

Minor revision

You described the utility of TM in PDTC and ATC in conclusions. However, you performed the study using an ATC cell line only. You had better modify conclusions based on the scope of your research.

Author Response

Reviewer 2

Comments and Suggestions for Authors

This study is interesting and informative. However, several issues should be solved.

Major comments

(Question 1)

To evaluate iodine-avidity after tunicamycin (TM) injection or the relationship between iodine uptake and therapeutic effects, iodine uptake (e.g., scan) in the tumor model should be compared between two groups (with or without TM injection). Or autoradiography might be possible if scan images were not available.

Response:

As per your comments, we additionally conducted a biodistribution study to evaluate radioiodine uptake in BHT101 tumors. As shown in Figure 1 for reviewer only, 0.02 mg/kg TM induced increased radioiodine avidity in BHT101 tumors. However, there was no increase in radioiodine avidity in vehicle-treated mice. Radioiodine uptake was also seen in the thyroid and stomach due to physiological reasons.

Figure 1 for reviewer only. When tumor mass was detectable by palpation and inspection, BHT101 tumor-bearing mice received 0.02 mg/kg TM via intraperitoneal injection twice a week for 2 weeks, followed by 1.85 MBq mCi 125I treatment intravenously. For biodistribution analysis, organs including tumors, liver, lung, heart, kidney, intestine, and others were removed, weighed, and tested for radioactivity using a gamma counter. The results were expressed as the percentage of injected dose per gram of tissue (%ID/g). **p<0.005, compared with vehicle.

(Question 2)

Increasing iodine avidity is very important in differentiated thyroid cancer (DTC) as well as ATC, because positive iodine uptake does not always result in a good therapeutic response in patients with DTC. Your study would be better if you showed the change of iodine uptake after TM injection and related molecular profiles in the DTC cell line and tumor model.

Response

We agreed with you. We have examined the effects of TM on radioiodine uptake, as well as iodide-handling genes expression of BCPAP as differentiated papillary thyroid cancer. As seen in Figure 2 for reviewer only, TM led to an increase in radioiodine avidity in BCPAP cells. KCLO4, as a NIS-specific inhibitor, completely reduced the enhanced radioiodine uptake to the basal level. Western blotting analysis showed the TM-induced up-regulation of NIS, TPO, and TSHR protein expression. These results suggested that TM is an effective agent for restoring radioiodine avidity in differentiated thyroid cancer.

Figure 2 for reviewer only. Effect of tunicamycin (TM) on iodide uptake in BCPAP cells. (A) BCPAP cells were treated with various doses of TM for 24 h, followed by iodide uptake assays. (B) To examine the inhibition of iodide uptake, ATC cells were treated with 2 µM TM and 300 µM KClO4 (NIS-specific inhibitor) for 24 h, and iodide uptake was assessed. **, p<0.005, ***, p<0.0005, compared to absence of TM (Iodide uptake assay). **, p<0.005 compared to TM (KCLO4 inhibition study). Data are the mean ± SD of three samples per group.

(Question 3)

Several drugs or agents (e.g., TKI, retinoic acid) have been studied for redifferentiation of thyroid cancer cells. So, you had better describe the effect of those agents in terms of redifferentiation with detailed manners and, if possible, compare your results with previous reports based on the degree of redifferentiation in the discussion section.

Response

Thank you for your valuable comments. We have previously examined the effects of TKI and retinoic acids as redifferentiation agents in anaplastic thyroid cancers. Even though these agents have been reported to be effective in differentiated thyroid cancer and non-thyroidal cancer (cf. MCF-7), they cannot induce the restoration of gene expression in thyroid-related genes and iodide uptake in anaplastic thyroid cancers. For example, as shown in Figure 3 for reviewer only, selumetinib as a potent and selective MEK inhibitor cannot upregulate NIS protein expression nor increase radioiodine uptake in BHT101 cells of anaplastic thyroid cancer, even though it effectively inhibits the phosphorylation of pERK1/2. Although we cannot address these phenomena, they will be associated with unknown and different signal pathways between differentiated thyroid cancer and anaplastic thyroid cancer.

Figure 3 for reviewer only. Effect of selumetinib on iodide uptake in BHT101 cells. (A) BHT101 cells were treated with various doses of selumetinib for 24 h. (A) Levels of NIS and p-ERK levels, as well as (B) iodide uptake avidity, was evaluated. N.S, not significant. Data are the mean ± SD of three samples per group.

Minor revision

(Question 4)

You described the utility of TM in PDTC and ATC in conclusions. However, you performed the study using an ATC cell line only. You had better modify conclusions based on the scope of your research.

Response

As you suggested, the description of TM utility in PDTC was deleted in the discussion and introduction sections to avoid confusion in understating the scope of our current research.

Round 2

Reviewer 2 Report

Please, merge new figures and legends to the manuscript or supplement (please do not reply answer (or figures) for reviewer only)

Your reply should be also merged into the text (including limitations).

Best regards.

Author Response

As reviewer suggested, new figures and legends was added to supporting information. And description for respective Figures was added in the result and discussion section of manuscript.